# Report on Vincristine-Producing Endophytic Fungus *Nigrospora zimmermanii* from Leaves of *Catharanthus roseus*

**DOI:** 10.3390/metabo12111119

**Published:** 2022-11-15

**Authors:** Kanchan Birat, Reem Binsuwaidan, Tariq Omar Siddiqi, Showkat Rasool Mir, Nawaf Alshammari, Mohd Adnan, Rahila Nazir, Bushra Ejaz, Moien Qadir Malik, Rikeshwer Prasad Dewangan, Syed Amir Ashraf, Bibhu Prasad Panda

**Affiliations:** 1Microbial and Pharmaceutical Biotechnology Laboratory, Department of Pharmacognosy & Phytochemistry, School of Pharmaceutical Education & Research, Jamia Hamdard, New Delhi 110062, India; 2Molecular Ecology Laboratory, Department of Botany, School of Chemical and Life Sciences, Jamia Hamdard, New Delhi 110062, India; 3Department of Pharmaceutical Sciences, College of Pharmacy, Princess Nourah bint Abdulrahman University, Riyadh P.O. Box 84428, Saudi Arabia; 4Phytopharmaceuticals Research Laboratory, School of Pharmaceutical Education & Research, Jamia Hamdard, New Delhi 110062, India; 5Department of Biology, College of Science, University of Hail, Hail P.O. Box 2440, Saudi Arabia; 6Department of Pharmaceutical Chemistry, School of Pharmaceutical Education & Research, Jamia Hamdard, New Delhi 110062, India; 7Department of Clinical Nutrition, College of Applied Medical Sciences, University of Hail, Hail P.O. Box 2440, Saudi Arabia

**Keywords:** vincristine, *Catharanthus roseus*, *Nigrospora zimmermanii*, endophyte, vinca alkaloids, anticancer compound, fungal metabolite

## Abstract

Vincristine is an anti-cancer compound and one of the most crucial vinca alkaloids produced by the medicinal plant *Catharanthus roseus* (L.) G. Don. (Apocynaceae). This plant is home to hundreds of endophytic microbes, which produce a variety of bioactive secondary metabolites that are known for their medicinal properties. In this study, we focused on isolating an endophytic fungus that could increase the yield of vincristine under laboratory conditions as an alternative to plant-mediated extraction of vincristine. The endophytic fungus *Nigrospora zimmermanii* (Apiosporaceae) was isolated from *Catharanthus roseus* and it was found to be producing the anticancer compound vincristine. It was identified using high-performance thin-layer chromatography by matching the Rf value and spectral data with the vincristine standard and mass spectrometry data and the reference molecule from the PubChem database. The generation study of this microbe showed that the production of vincristine in the parent fungus was at its maximum, i.e., 5.344 µg/mL, while it was slightly reduced in subsequent generations. A colonization study was also performed and it showed that the fungus *N. zimmermanii* was able to re-infect the plant *Catharanthus roseus* after 20 days of inoculation. The colonization study showed that *N. zimmernanii* could infect the plant after isolation. This method is an efficient and easy way to obtain a high yield of vincristine, as compared to plant-mediated production.

## 1. Introduction

Endophytes are microorganisms that inhabit different parts of plants, generally without causing any apparent symptoms of disease. Endophytes show various associations with their host plants, such as symbiosis, pathogenesis, antagonistic relationships, etc., and may promote plant growth, counter plant stress conditions, and even elicit plant defense against any pathogenic attack [1]. They increase the defense mechanism of their host plants by producing similar secondary metabolites, and therefore they are crucial for ex-planta production of these compounds [2]. These microbes are of immense importance in nature because they are known to produce bioactive secondary metabolites, many of which may be used as antidiabetic, antibacterial, antifungal, and anticancer compounds [3,4]. Interestingly, these compounds are identical to those that are originally produced inside the host plants and are of great importance in the medical industry. Frequently, these endophytes also act as elicitors for in vivo enhancement of the bioactive secondary metabolites produced by their host plants, thereby increasing the production rate of such compounds [5]. New avenues are being explored to make use of these endophytes to their fullest. One of the methods is to grow these endophytes on a large scale in controlled conditions with appropriate supplements to obtain the desired product in huge amounts, which might help in decreasing the marketing cost as well as increasing the ease of production.

*Catharanthus roseus* (L.) G. Don (Appendix A) is one such plant that is home to hundreds of such endophytic microbes. It is also called Vinca rosea, Madagaskar periwinkle, Cape periwinkle, old maid, Sadabahar, etc. It belongs to the family Apocynaceae, whose members are most commonly used for their medicinal properties. *C. roseus* is traditionally used to treat diabetes. This plant has anticancer, antidiabetic, anti-Alzheimer, and wound-healing properties [3,6]. It is also known to produce a variety of terpenoid indole alkaloids such as vincristine, vinblastine, catharanthine, ajmalicine, vindoline, etc. which are of immense importance in the medical industry [7].

Vincristine and vinblastine are potent anticancer compounds that are known to perform their activities by arresting the cell in the mitosis stage [8]. Tubulin protein inside a cell polymerizes into long chains or filaments to form microtubules that help in chromosomal movement during cell division. Vincristine, which is composed of two multi-rings, namely vindoline and catharanthine, binds to the β-tubulin unit, preventing its polymerization and formation of spindle microtubules, thereby arresting cell division. [9,10]. Vincristine is used in the form of sulfate salt injections to treat Hodgkin’s and non-Hodgkin’s lymphomas, Wilkins’s tumors, brain tumors, acute lymphocytic leukemia, thyroid cancer, neuroblastoma, etc. [11,12]. Vincristine is also reported to be used for the treatment of non-malignant hematologic disorders [13].

Nigrospora species are ubiquitous and grow rapidly on potato dextrose agar by producing wooly colonies. The colonies of *N. zimmermanii* are initially white but, after 4–5 days, turn gray to black. It produces round conidia that are coffee brown and takes the stain lactophenol cotton blue. In the present study, an endophytic fungus, *Nigrospora zimmermanii*, belonging to the family Apiosporaceae (Xylariales, Sordariomycetes), was isolated for the first time from the plant *C. roseus.* Different species of *Nigrospora*, which is generally known as a plant pathogen, have the capability to produce diversified secondary metabolites [14]. In 2017, *Nigrospora sphaerica* isolated from *C. roseus* was reported to produce vincristine in a study by Ayob et al. [15]. New avenues are being explored to make use of these endophytes to their fullest potential. One of the methods is to grow these endophytes on a large scale in controlled conditions with appropriate supplements to obtain the desired products in huge amounts in a much faster and more efficient manner, which might help in decreasing the marketing cost as well as increasing the ease of production of plant secondary metabolites by microbial cells.

## 2. Materials and Methods

### 2.1. Endophytic Microbe Isolation

Healthy plant samples (leaves, stems, and roots) were collected in sterile bags using sterile tools from various regions of India, including Delhi (latitude and longitude coordinates are: 28.51559, 77.25078), Haryana (latitude and longitude coordinates are: 28.48759, 77.28667), and Rajasthan (latitude and longitude coordinates are: 27.55329, 76.63419). The stem and root cuttings and whole leaves with petioles were subjected to surface sterilization. In the laminar hood, the leaves were immersed in 70% ethanol for 30 s and the stem and root cuttings were immersed for 40 s. The explants were washed twice with autoclaved water and then immersed in 0.01% mercuric chloride. The leaves were kept there for 50 s and the stem and root cuttings were kept for 70 s. These were washed again three times with autoclaved water to ensure that the mercuric chloride was washed off thoroughly. The residual water on the explants was soaked on sterile blotting paper. The explants were then cut into pieces with approximate dimensions of 0.5 cm diameter (root and stem) and 0.5 × 2 cm (leaves with midrib) using a sterile scalpel. These pieces were placed on Petri plates poured with potato dextrose agar (PDA) and incubated at 26 ± 2 °C. After 10 days, individual colonies were transferred to different PDA plates to obtain pure colonies of the endophytic fungi. These were collected in PDA slants as well, and stored at 4 °C for future use. The endophytic fungi so obtained were then grown in 100 mL of potato dextrose broth (PDB). To start the suspension culture, a 5 mm diameter agar block was taken from every Petri plate in which the fungal culture had been established. The culture conditions used for the suspension culture were 26 ± 2 °C and 230 rpm. After 8 days of incubation, the mycelia from the filtrate were separated using a muslin cloth and Whatman filter paper number 1. The filtrate was measured and the obtained mycelia were weighed to obtain the wet weight. The filtrate was then concentrated to half its volume using a rotary evaporator at 90 °C. The concentrated filtrate was collected in a falcon tube and an equal amount of ethyl acetate was added to it. After vortexing the mixture, it was left still for 2 min. The mixture was separated by keeping it in a centrifuge at 5000 rpm for 15 min and the organic layer was collected in a separate falcon tube. The remaining solution was again filled with equal amounts of ethyl acetate and the process was repeated three times to obtain the maximum of the compound. The obtained ethyl acetate extract was then fully evaporated and reconstituted with 1.5 mL of methanol to make a methanolic extract.

The obtained mycelia separated from the filterate were transferred into a glass tube. Dimethy sulfoxide (DMSO) was first pre-heated in an oven at 60 °C for 30 min and then it was transferred to the mycelia in a 1:1 *w*/*v* ratio and sonicated for 15 min to rupture the fungal cells. The sonicated material was then filtered using a muslin cloth and Whatman filter paper number 1 to obtain the filtrate. An equal amount of ethyl acetate was added to the filtrate, mixed properly by vortexing, and the organic layer was obtained by separating the mixture in a centrifuge at 5000× *g* for 15 min. The process was repeated thrice and the obtained filtrate was evaporated completely and reconstituted with 1.5 mL of methanol. These methanolic extracts were then used for further analysis.

### 2.2. HPTLC Conditions for Qualitative and Quantitative Analysis

The HPTLC silica gel 60 F254 (Merck) plates were pre-washed with methanol by dipping them in an HPTLC chamber filled with 20 mL of methanol and letting the solvent overrun to remove any impurities, and then the plate was activated for 10 min at 120 °C in an oven. Toluene–methanol–diethylamine was used as a mobile phase in a ratio of 8.75:0.75:0.5 [16]. A separate chamber with its walls covered with tissue paper was allowed to mix with the mobile phase for 45 min to one hour at room temperature. The spots of 6mm band length were marked using a Linomat 5 auto-sampler and were captured under UV light. The solvent was allowed to rise over the HPTLC plate by 90% in the Camag twin-trough chamber saturated with the mobile phase. The plate was taken out and kept in an upright position for air-drying. It was then kept in the oven for 30 s at 75 °C to evaporate any traces of the mobile phase and scanned at 300 nm. The standard for vincristine was obtained from Sigma Aldrich, Bangaluru, India and a stock solution of 1 mg/mL in methanol was prepared, from which different dilutions were made. The dilutions used were 200 ng/mL, 400 ng/mL, 600 ng/mL, 800 ng/mL, and 1000 ng/mL for HPTLC analysis.

### 2.3. UPLC-MS Analysis Conditions

UPLC-MS analysis was performed by using a Waters Acquity UPLC-MS/MS instrument (Waters Corporation, Milford, MA, USA) with a binary solvent manager system. All the data were acquired using MassLynx 4.1 software using full MS scanning mode. Chromatographic separation was achieved on an ACQUITY UPLC^®^ BEH C18, 1.7 µm particle size, (2.1 × 100) mm-column maintained at 40 °C temperature throughout the run. The mobile phases consisted of water containing 0.1% formic acid (mobile phase A) and acetonitrile containing 0.1% formic acid (mobile phase B). The gradient elution was performed at a flow rate of 0.4 mL/min and the injection volume was 10 µL for each sample and the total run time was 16 min. Initially, A and B were isocratic in the ratio of 90:10 for 1 min, which was changed linearly to a ratio of 60:40 in the next 2 to 4 min. After 5 to 10 min, the A:B gradient was changed to 40:60, and at 10–13 min, the ratio was changed to 10:90, and finally, at 15 min, the ratio became 0:100, which was further reversed to 90:10 in 16 min.

MS detection was conducted by monitoring the calculated mass of vincristine [(M + H)^+^ = 825.4]. Parameters for the same were as follows: capillary voltage, 3.50 kV; cone voltage, 40 V; source temperature, 110 °C; desolvation temperature, 350 °C; cone gas flow, 50 L/h; desolvation gas flow, 800 L/h. The MS scanning ranges were applied between 100–1000 Da in electrospray positive-ion mode.

### 2.4. Microbe Identification

DNA extraction using the CTAB method proceeded as follows: The fungal strain was picked up from the slants previously prepared and placed in a mortar and homogenized with 500 µL of CTAB (Cetyltrimethylammonium bromide). The homogenate was transferred to a microcentrifuge tube. An equal volume of phenol:chloroform:isoamyl alcohol (25:24:1) was added to the tube and mixed well by gently shaking the tube. The sample was centrifuged at room temperature at 14,000 rpm for 15 min. The upper aqueous phase was collected in a new tube and an equal volume of chloroform:isoamyl alcohol (24:1) was added and mixed. The upper aqueous phase obtained after centrifuging at room temperature for 10 min at 14,000 rpm, was transferred to a new tube. The DNA was precipitated from the solution by adding 0.1 µL volumes of 3 M sodium acetate (pH 7.0) and 0.7 µL of isopropanol. After 15 min of incubation at room temperature, the tube was centrifuged at 4 °C for 15 min at 14,000× *g* rpm. The DNA pellet was washed twice with 500 µL of 70% ethanol and then briefly with 100% ethanol and air-dried. The DNA was dissolved in 20 µL TE (Tris-Cl 10 mM pH 8.0, EDTA 1 mM) buffer. To remove the RNA, 5 µL of DNase-free RNase A (10 mg/mL) was added to the DNA. The DNA fragments are amplified using high fidelity.

#### 2.4.1. PCR Polymerase

PCR conditions were as follows: Amplification of the nuclear internal transcribed spacers (ITS) region of the 18S rRNA gene was performed using the universal primers: ITS1 (5′-TCCGTAGGTGAACCTGCGG-3′) and ITS4 (5′-TCCTCCGCTTATTGATATGC-3′) [17]. Polymerase chain reaction (PCR) was performed in a Master cycler^®^ Thermocycler (Eppendorf, Germany) with a total reaction volume of 50 µL made up of 1 µL DNA, 2 µL ITS forward primer, 2 µL ITS reverse primer, 4 µL dNTPs (2.5 mM each), 10 µL 10× Taq DNA polymerase assay buffer, 1 µL Taq DNA polymerase enzyme (3 U/µL), and 30 µL water. Initial denaturation at 94 °C for 3 min was followed by 35 cycles of denaturation at 94 °C for 1 min, annealing at 50 °C for 1 min, and extension at 72 °C for 2 min. The final extension was carried out at 72 °C for 7 min. Amplified PCR products were separated in 1% agarose gel containing ethidium bromide in 0.5 μg/mL concentration (for detection of bands under UV) in Tris–borate–EDTA (TBE) buffer. The purified PCR product was used for sequencing.

#### 2.4.2. Sequencing and PCR Conditions

Sequencing was carried out in an ABI 3130 genetic analyzer using 10 µL of sequencing reaction and 25 cycles of PCR with initial denaturation of 96 °C for 5 min, denaturation at 96 °C for 30 s, hybridization at 50 °C for 30 s, and elongation at 60 °C for 1.30 min.

#### 2.4.3. Phylogenetic Tree

A phylogenetic tree was built using Phylogenetic Tree Builder, which uses sequences aligned with the System Software aligner. A distance matrix was generated using the Jukes–Cantor corrected distance model. While generating the distance matrix, only alignment model positions were used, alignment inserts were ignored, and the minimum comparable position was 200. The tree was created using Weighbor with alphabet size 4 and length size 1000 [18,19].

### 2.5. Generation Study

The identified parent fungus was cultured for six more generations on PDA media by inoculating subsequent plates with the preceding generation of the fungus. All seven generations of the obtained fungus were then subjected to suspension cultures in 100 mL of potato dextrose broth (PDB) and the process was repeated to obtain the 1.5 mL of methanolic extract. These samples were used for the quantification of vincristine content by HPTLC.

### 2.6. Colonization Study

The funagal inoculum was prepared by growing the fungus in PDB media at 26 ± 2 °C and 230 rpm for 8 days. A fresh plant was re-infected through the stem by the desired fungal inoculum (2 mL; 6 × 10^4^ cfu per mL). A control experiment was run before the re-infection to check for any traces of endophyte already present in the plant. The region that was chosen for re-infection was somewhere in the middle portion of the plant body, away from the apical meristem and near the budding sight of leaves, where a longitudinal wound was made through the diameter of the stem, taking into consideration that the phloem and xylem tissue should be exposed to the fungal inoculum. The wound was then covered with muslin cloth and the fungus was allowed to spread inside the plant for 20 days. No morphological change was seen in the plant and the healthy leaves were collected from near the infected area. The presence of endophytic fungus in the infected plant was checked by isolating the endophytic fungus from the leaf sample. The fungal mycelia is stained using lactophenol cotton blue and safranin to obtain adequate observation of the colonization under the microscope.

### 2.7. Phenotypic Identification

For the morphological analysis of the fungal strain, a slide culture was performed. For this, a small agar block was placed on a clean sterile slide and the agar block was covered with a coverslip. The sides of the agar block were then inoculated with the desired fungal strain using an inoculating loop. The slide was placed in a Petri plate containing a wet sterile blotting sheet to keep the compartment moist. This plate was incubated at 26 ± 2 °C. After 7 days the agar block was discarded and the coverslip was placed on a fresh clean slide after adding lactophenol cotton blue to the fungal mycelia obtained on the coverslip. The slide was placed under a microscope to observe the morphology of the fungal strain.

## 3. Results

### 3.1. Microbe Isolation

A total of 37 endophytic fungi were isolated from the root, stem, and leaves of *Catharanthus roseus*. Only one fungus (Figure 1a,b), isolated from the leaf, was found to be producing the anticancer compound vincristine. The colonies were observed for 10 days and initially were found to be white and cottony for 4–5 days, but the microbe gradually changed its color to gray-black when observed from the front as well as from the back of the Petri plate; no set pattern of the colony was observed, and the mycelia were scattered all through the plate. In the slide culture, the fungal hyphae, hypha with conidiophores, and conidia were visible under the light microscope at 400× magnification (Figure 1c,d). Micrometry observation shows the diameter of the hyphae was from 2 to 4 µm; conidiophores were mostly reduced to conidiogenous cells, with black and globose-shaped conidia measuring 11–14 µm in size.

### 3.2. Colonization and Generation Study

The eight-day suspension culture of the parent generation showed a whitish fungal growth (Appendix A). However, a change in color from gray to black was observed in subsequent generations in the eight-day suspension culture (Appendix A–g). The plant looked healthy even after 45 days of re-infection with endophytic fungus (Figure 2). The colonization study showed that the fungal strain was able to re-infect the plant tissue. The mycelia and spores inside the leaf sample were visible under the light microscope at 400× magnification (Appendix A–c).

### 3.3. Microbe Identification

The band observed for the ITS region was ~750 bp (Appendix A). The fungal strain was confirmed by sequence analysis and the formation of a phylogenetic tree (Figure 3). Sequence analysis of the ITS region of the 18S rRNA gene showed that the microbe was *Nigrospora zimmermanii*. The next closest homolog was found to be *Nigrospora pyriformis* CGMCC 3.18122 ITS region (Appendix A; Appendix A).

### 3.4. Qualitative Analysis of Metabolites of Endophytic Fungus

The HPTLC, LCMS, and mass spectrometry data showed that the anticancer compound vincristine was being produced by the fungus in its extracellular filtrate but the intracellular filtrate was not giving any data about the presence of vincristine. The Rf for vincristine standard was recorded to be 0.17 (Figure 4a–h). The regression via area was recorded to be 0.98 (Appendix A). UPLC-MS data showed a similarity between the chromatogram of the vincristine standard and the vincristine molecule present in the sample (Figure 5). The molecular weight (*m*/*z*) recorded for the sample was calculated as [M + 2H]^2+^ = 413.2458 (Figure 6) through mass spectrometry, which was similar to that of the molecular weight of vincristine in the PubChem database.

### 3.5. Quantitative Analysis of Metabolites of Endophytic Fungus

The quantity of vincristine produced by the isolated endophytic fungus was measured through HPTLC analysis. It was observed that the compound was present in the extracellular filterate of the fungal culture, and it was found to be maximum in the methanolic extract of the parent fungus filtrate (Generation 1) isolated from the *C.* roseus, i.e., 5.344 µg/mL. However, an irregular pattern of decrease in vincristine production in subsequent generations was observed (Figure 4c–h). The quantity of vincristine produced by Generation 2 was found to be 1.517 µg/mL, 2.647 µg/mL in Generation 3, 1.066 µg/mL in Generation 4, 1.150 µg/mL in Generation 5, 2.506 µg/mL in Generation 6, and it was absent in Generation 7 (Table 1) (Appendix A). The spectral data of samples also matched with the spectral data of vincristine in standard, which confirms that the product was vincristine. UV spectrum showed the λ max for vincristine at 300 nm (Figure 7).

## 4. Discussion

*C. roseus* is a plant home to numerous endophytic fungi, which are of immense importance for obtaining various medicinally important compounds. Vincristine is an anti-cancer compound that is naturally produced inside the plant *C. roseus*. The in-planta yield of vincristine is approximately 0.0005% of the dry weight in *C. roseus*; therefore alternative methods for the production of vincristine are being explored [20]. One of the alternatives is the endophytes inhabiting *C. roseus* and other medicinal plants, which adapt to live in association with their host plant and make such changes in their genetic makeup that they can produce similar compounds to those of their host [21,22]. Such endophytes are of immense importance for obtaining the desired compounds on a large scale [23]. However, one most crucial observation in this study is that the production of these compounds in endophytic fungal culture decreases irregularly in subsequent generations. A probable reason for this decrease in the production of vincristine could be the change in genetic makeup, because most endophytes produce secondary metabolites in association with their host plants. The absence of an intermediate compound and enzyme availability in the culture media, which might have been available while growing inside the host plant, might also be a reason for this change [24]. A colonization experiment in the present study showed that the fungus *N. zimmermanii* could re-enter the plant tissue after being isolated in the laboratory. Very old plants, as well as young plants, were not used for re-infection because any other traces of endophytic fungal infection could be found in older plants, and the very young plant would be replicating too rapidly, preventing the fungal mycelia from multiplying efficiently inside the plant. Therefore, a middle-aged plant was selected for this experiment.

It would be interesting to note that if the fungus stops producing vincristine at any point in time due to any of the above-mentioned reasons, vincristine production could be revived by re-infecting the plant with this fungus. Further studies are required to be carried out to support this idea. However, the ability to grow rapidly under a controlled environment gives fungi an edge over any other source for the large-scale production of anticancer compounds such as vincristine, which is an expensive drug for treating various types of cancers. Different studies show that a plethora of endophytic fungi that produce anticancer compounds has been isolated, such as camptothecin, taxol, paclitaxel, podophyllotoxin, hypericin, and many more [25]. This study opens up a way for large-scale production of vincristine by culturing the endophytic fungus *N. zimmermanii* in a bioreactor, thereby reducing the marketing as well as the production cost of this anticancer compound.

## 5. Conclusions

This study showed that an endophytic fungus, *N. zimmermanii*, isolated from *C. roseus* produced the anticancer compound vincristine. A generation study was performed to check for the stability of the compound in subsequent generations under laboratory conditions. The microbe was found to be producing vincristine until the sixth generation, with a slight decrease in production. The colonization study showed that *N. zimmernanii* could infect the plant after isolation. This method is efficient and easy for obtaining a high yield of vincristine, as compared to plant-mediated production. However, further study may be required for the stability of compounds in subsequent generations of the isolated fungus.

## Figures and Tables

**Figure 1 metabolites-12-01119-f001:**
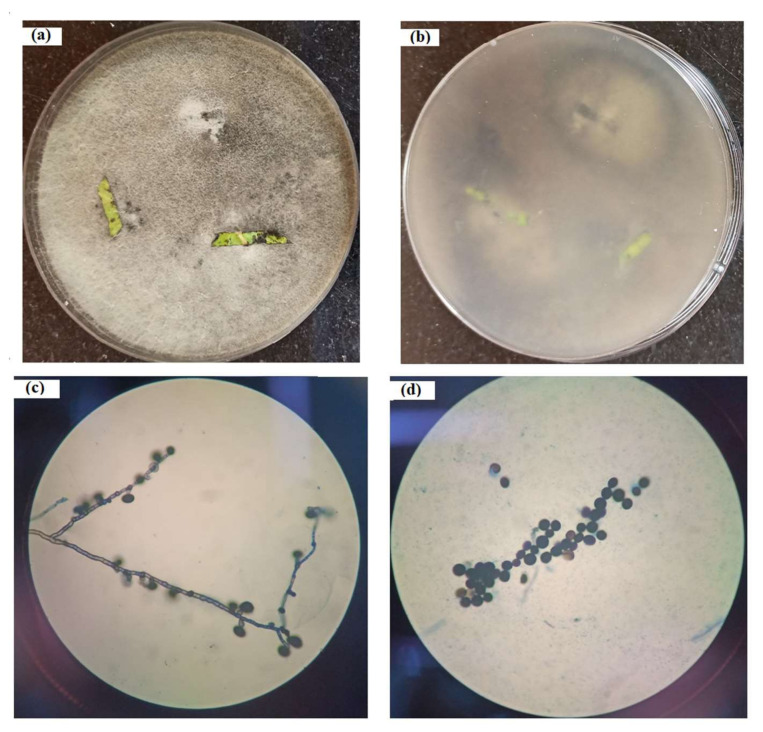
Isolated fungus *Nigrospora zimmermanii* on PDA media, showing (**a**) top view; (**b**) back view, microscopy of the isolated fungus *Nigrospora zimmermanii*; (**c**) hypha with conidiophores; (**d**) conidia.

**Figure 2 metabolites-12-01119-f002:**
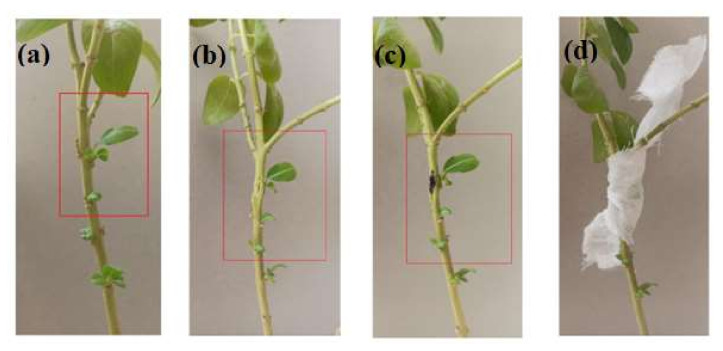
Re-infection of endophytic fungus in a healthy *Catharanthus roseus* plant, showing (**a**) chosen site for infection; (**b**) creation of wound; (**c**) inoculation of desired endophytic fungus; (**d**) site of infection covered with a muslin cloth.

**Figure 3 metabolites-12-01119-f003:**
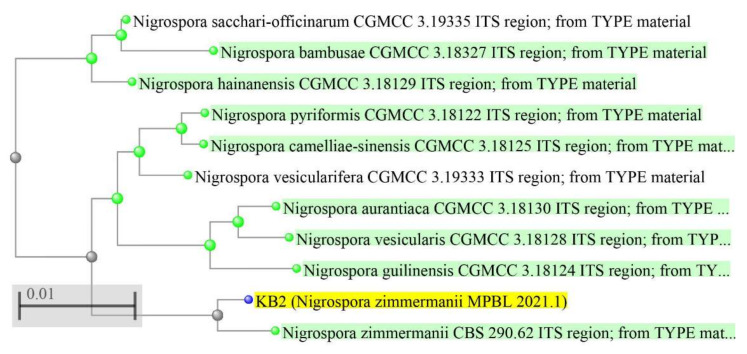
Phylogenetic tree with most recent common ancestor (green dot and green highlights) and the ultimate common ancestor (gray dot) of the desired isolated fungus *Nigrospora zimmermanii* (blue dot and yellow highlights).

**Figure 4 metabolites-12-01119-f004:**
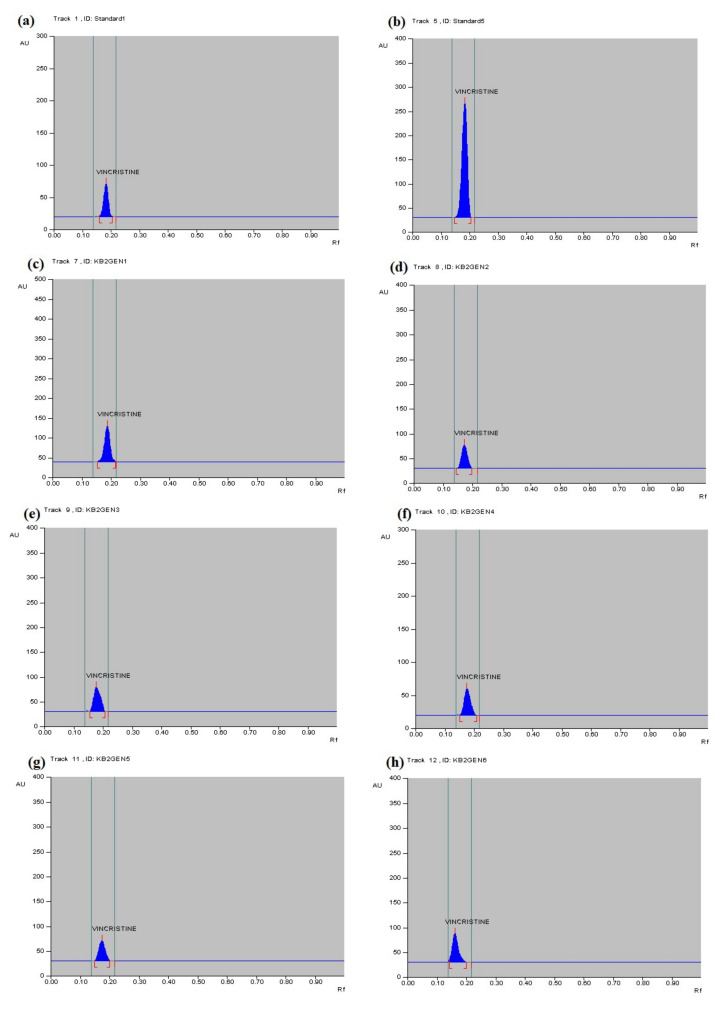
HPTLC peaks for vincristine, showing (**a**) standard at 200 ng/mL; (**b**) standard at 1 mg/mL; (**c**) generation 1 at 300 ng/mL; (**d**) generation 2 at 300 ng/mL; (**e**) generation 3 at 300 ng/mL; (**f**) generation 4 at 300 ng/mL; (**g**) generation 5 at 300 ng/mL; and (**h**) generation 6 at 300 ng/mL.

**Figure 5 metabolites-12-01119-f005:**
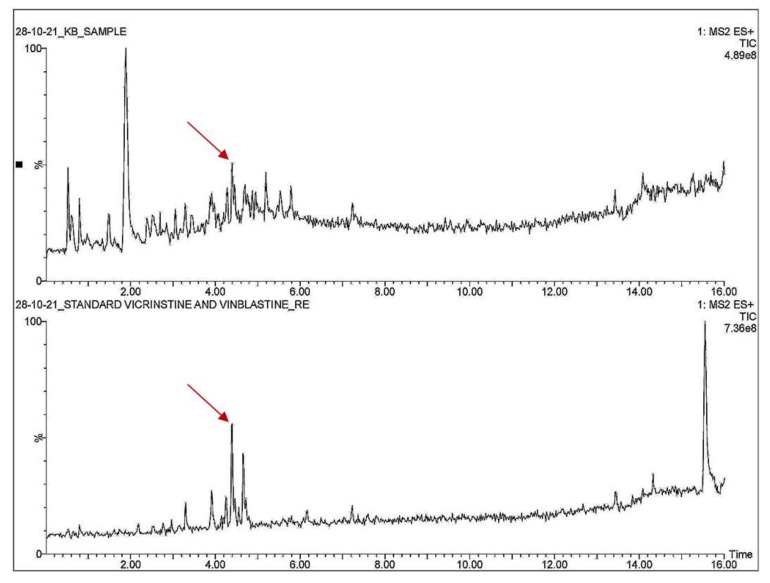
LC–MS chromatogram for vincristine in sample and vincristine standard.

**Figure 6 metabolites-12-01119-f006:**
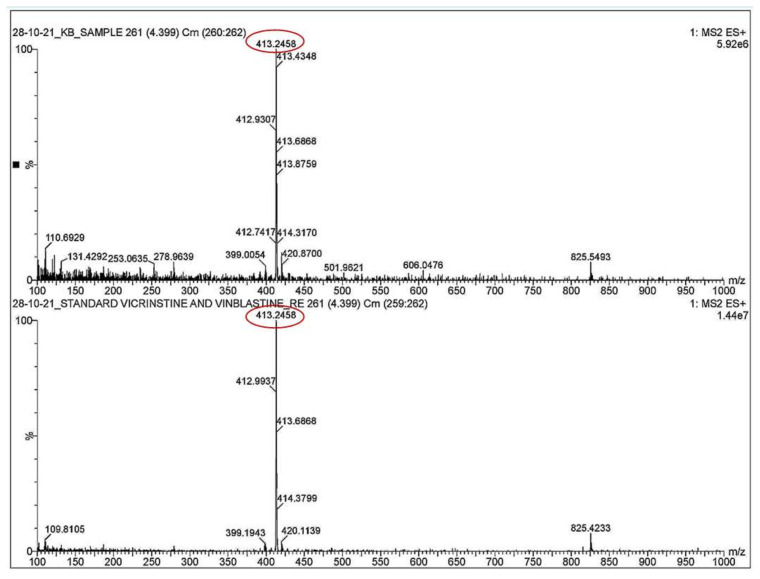
Mass spectrometry for vincristine in sample and vincristine standard showing m/z [M+2H]^2+^ = 413.2458.

**Figure 7 metabolites-12-01119-f007:**
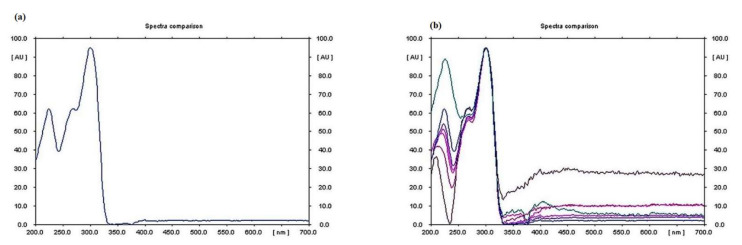
Spectral data for vincristine, showing (**a**) λ max at 300 nm in standard; (**b**) λ max at 300 nm in samples.

**Table 1 metabolites-12-01119-t001:** Comparative data of generation study of desired fungal strain *Nigrospora zimmermanii*.

S. No.	Generation	Color of Colony	Vincristine Content Produced in Extracellular Fungal Methanolic Extract (µg/mL)
1	1st	White	5.344
2	2nd	Pale white to grey	1.517
3	3rd	Grey	2.647
4	4th	Grey	1.066
5	5th	Brown	1.150
6	6th	Dark brown	2.506
7	7th	Black	Nil

## Data Availability

All the data supporting the findings of this study are included in this article and the supplementary files. Any other data related to this study are also available on request from the corresponding author.

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
