# Peer review of "Report on Vincristine-Producing Endophytic Fungus Nigrospora zimmermanii from Leaves of Catharanthus roseus"

_metabolites, 2022, doi:10.3390/metabo12111119_

Round 1

Reviewer 1 Report

MANUSCRIPT DETAILS

Ms. Ref. No. metabolites-2000690

Title: Report on Vincristine producing endophytic fungus Nigrospora zimmermanii from leaves of Catharanthus roseus

Article Type: Research Article

Journal: Metabolites

GENERAL COMMENTS

The interest in this manuscript is significant enough to merit publication.

My recommendation on submitted manuscript to the Metabolites is to be accepted after minor revisions.

The comments and questions provided below may help the authors to put the manuscript into better appropriate form for publication.

SPECIFIC COMMENTS

- Abstract

·       Authors should clarify the aim of study in abstract and at end of introduction

- Material and Methods

·       Authors should add the conditions of samples transfer 

·       Line 228 Authors should change subtitle of Microscopy to phenotypic identification  

- Result and discussion

·       Line 261 Authors should add Accession No. of KB-2 in fig. 2

·       Line 296 HPTLC peaks is not clear  

·       Authors should discuss Quantitative analysis of metabolites of endophytic fungus in detail

Therefore, I think this paper is acceptable for publication with minor revision.

Author Response

Reviewer 1

Ms. Ref. No. metabolites-2000690

Title: Report on Vincristine producing endophytic fungus Nigrospora zimmermanii from leaves of Catharanthus roseus

Article Type: Research Article

Journal: Metabolites

GENERAL COMMENTS

The interest in this manuscript is significant enough to merit publication.

My recommendation on submitted manuscript to the Metabolites is to be accepted after minor revisions.

The comments and questions provided below may help the authors to put the manuscript into better appropriate form for publication.

SPECIFIC COMMENTS

- Abstract

Authors should clarify the aim of study in abstract and at end of introduction

Ans: the aim of study in abstract is added in the revised manuscript.  

- Material and Methods

Authors should add the conditions of samples transfer 

Ans: The condition of sample transfer is revised in the manuscript. 

Line 228 Authors should change subtitle of Microscopy to phenotypic identification  

Ans: Correction is made as per the suggestion.

- Result and discussion

Line 261 Authors should add Accession No. of KB-2 in fig. 2

Ans: Accession No. of KB-2 in fig. 2 is added in the revised manuscript.

Line 296 HPTLC peaks is not clear  

Ans: HPTLC Peaks are revised in the manuscript with clarity. 

Authors should discuss Quantitative analysis of metabolites of endophytic fungus in detail

Ans: Quantitative analysis of metabolites is discussed in detail in the revised manuscript.

All the authors are thankful to the reviewer for evaluating the article and giving valuable suggestions to improve the article.

Dr. Bibhu Prasad Panda

Email: bppanda@jamiahamdard.ac.in

Reviewer 2 Report

The study by Birat et al. reports the isolation of the endophytic fungus Nigrospora zimmermanii from the plant Catharanthus roseus. The fungus was shown to produce the anticancer compound vincristine until 7th generation, with a maximum production in generation 1. The study is in general well done, but need some extensive revision mostly in the way the results are presented. There are irrelevant figures that should be shown as supplementary information, and vice versa, and others that should be combined into one.

Major commets:

 It is not clear, at least to me, to what extent the anticancer compound is produced by the plant, the fungus, or both. An experiment that would be lacking to confirm this point is to quantify vincristine from plants before the fungus was re-inoculated and in plants with the presence of the first re-inoculated generation.

Minor comments:

Line 25: Why “important”?

Line 62: this is science, please delete “blessed”.

Line 76 to 78: How the fungus is first isolate if you are citing as previous information? (#14)

Line 79 vs Ref.#15 in bibliography: Ayub or Ayob?

Information of lines 81-83 should be placed before, and finish the introduction with a more general statement.

Section 2.1: please georeference the collection sites.

Line 55: rewrite “…dimensions of 0.5 cm diameter (root and stem) and 0.5 cm…

Line 113: Only one compound was obtained?

Line 117: sonication causes rupture of fungal cells? Or just affects membrane permeability? For internal metabolites release is the same but please clarify.

Fig. 1: put only “A” and “B” in panels. This figure does not show the variety of fungi described in text, nor does it show that a single fungus produce vincristine.

Fig. 2: please detail why some species are highlighted, colors of dots, etc.

Table 1 can be placed as supplementary table.

Line 274: why the colors change?

Fig. S2: genomic DNA has a band of only 525 bp in agarose gel?

Fig. 1 and 3 can be combined in only one figure.

Why vincristine has a peak at 300 nm?

Line 332: delete “wonderful”

First statement in conclusion is repeated.

Author Response

Reviewers 2

Comments and Suggestions for Authors

The study by Birat et al. reports the isolation of the endophytic fungus Nigrospora zimmermanii from the plant Catharanthus roseus. The fungus was shown to produce the anticancer compound vincristine until 7th generation, with a maximum production in generation 1. The study is in general well done, but need some extensive revision mostly in the way the results are presented. There are irrelevant figures that should be shown as supplementary information, and vice versa, and others that should be combined into one.

Major comments:

It is not clear, at least to me, to what extent the anticancer compound is produced by the plant, the fungus, or both. An experiment that would be lacking to confirm this point is to quantify vincristine from plants before the fungus was re-inoculated and in plants with the presence of the first re-inoculated generation.

Answer: The literature review specifies that the amount of vicristine obtained by the plant itself is very less. For obtaining 100g of crude Vincristine requires about 2 tons of Catharanthus roseus leaves. The process of obtaining vincristine from plant is a cumbersome process too. The main aim of the study was to obtain a rapidly replicating Vincristine producing microbe under controlled environment so as to cultivate it in bulk and thereby obtaining high yield of vincristine from the desired microbe in an easy and efficient way. By the experiment it is evident that the microbe can be cultured in the laboratory there by producing Vincristine more than what can be obtained by the plant itself.

Our experiments showed that: the endophyte is able to produce 5.344 µg/ml in extra-cellular.  i.e. one liter of culture will produce 5.3mg of pure vincristine in short period as compared to plant. We agree that the reinfection study was performed only to know about the re-infection capability of the endophyte but not related to metabolites production capability by the infected plant.

Minor comments:

Line 25: Why “important”?

Ans: Endophytic fungus are important because of their capability to produce the same metabolites as that of plant. However, they stop producing the metabolites after certain generation once they grow outside the plant tissue. The exact mechanism why this happens is still a point to understand.

Line 62: this is science, please delete “blessed”.

Ans: The is corrected as per the suggestion.

Line 76 to 78: How the fungus is first isolate if you are citing as previous information? (#14)

Ans: Wang et al 2014, reported about different Species of Nigrospora, the citation is corrected and revised.

Line 79 vs Ref.#15 in bibliography: Ayub or Ayob?

Ans: The citation is corrected and revised.

Information of lines 81-83 should be placed before, and finish the introduction with a more general statement.

Ans: The is corrected as per the suggestion.

Section 2.1: please geo reference the collection sites.

Ans: Geo references is added in the revised manuscript

Line 55: rewrite “…dimensions of 0.5 cm diameter (root and stem) and 0.5 cm…

Ans:  Revised as per the suggestion.

Line 113: Only one compound was obtained?

Answer: In the extract more than one compounds were obtained.  The target compound for the present study was vincristine hence the study was conducted using the specific standard avoiding any other compound detection.

Line 117: sonication causes rupture of fungal cells? Or just affects membrane permeability? For internal metabolites release is the same but please clarify.

Answer: The purpose of sonication was to rupture the cell and release compounds, if any, that were produced by the fungal cell.

Fig. 1: put only “A” and “B” in panels. This figure does not show the variety of fungi described in text, nor does it show that a single fungus produce vincristine.

Answer: A total of 37 endophytic fungi were isolated from the root, stem, and leaves of Catharanthus roseus. There is only one fungal strain in the whole plate which was isolated from C. roseus leaves. After conducting suspension culture and qualitative analysis of compounds produced by this fugus did we know that it was producing Vincristine.

Fig. 2: please detail why some species are highlighted, colors of dots, etc.

Answer: Color of dots is only for the distinction in the phylogenetic tree with most recent common ancestor (having green dot) and the ultimate common ancestor (having gray dot) of the desired fungus (represented with blue dot).

Table 1 can be placed as supplementary table.

Ans:  Revised as per the suggestion.

Line 274: why the colors change?

Answer: By observing the suspension culture we can rule out that the change in color in subsequent generations is either due to production of secondary metabolites by the fungal strain under laboratory conditions or due to change in color of spores or mycelia because of growing it in artificial culture media for many generations. [According to a book (Annals of Botany by KRIPA RAM MOHENDRA)]. However further research is required for these answers.

Fig. S2: genomic DNA has a band of only 525 bp in agarose gel?

Ans: The gDNA is larger size, the concentration of the DNA is found to be 54.4ng/µl. Corrected gel is added in supplementary figure with all the ladder size data.

Fig. 1 and 3 can be combined in only one figure.

Answer: The purpose of keeping them separate was to align the figures with text and easy access of information.

Why vincristine has a peak at 300 nm?

Answer:  In our TLC experiment it is observed at 300nm, Different concentrations of vincristine standard were used for quantitative analysis and the peaks are according to the decreased or increased concentration. Fig 4(a) shows the lowest concentration of standard used and Fig 4(b) shows the highest concentration of standard used. The higher the concentration, the higher the peak.

Vincristine in methanol is showing UV max at: 218,252, 285, 293 nm (Source PubChem)

Ref: O'Neil, M.J. (ed.). The Merck Index - An Encyclopedia of Chemicals, Drugs, and Biologicals. Whitehouse Station, NJ: Merck and Co., Inc., 2006., p. 171

In HPTLC: The max UV for scanning the TLC plate for vincristine is 307nm 

Ref: Abid, K.; Sayeed, A.; Jalees, A.F.; Kishwarb, S. Simultaneous determination of Vincristine and Vinblastine in Vinca 443 rosea leaves by High Performance Thin Layer Chromatography. Int. J. Drug Dev. & Res. 2013, 5 (3), 341-348.

In HPLC-UV: The Max UV for vincristine is 297nm 

Ref: Golpayegani MR, Akramipour R, Gheini S, Amini MV, Fattahi F, Mohebbi A, Fattahi N. Sensitive determination of vincristine in plasma of children with leukaemia using vortex-assisted dispersive liquid-liquid microextraction based on hydrophobic deep eutectic solvent. RSC Adv. 2022 Jan 26;12(6):3611-3617.

Line 332: delete “wonderful”

Ans:  Revised as per the suggestion

First statement in conclusion is repeated.

Ans:  Revised as per the suggestion

All the authors are thankful to the reviewer for evaluating the article and giving valuable suggestions to improve the article.

Dr. Bibhu Prasad Panda

Email: bppanda@jamiahamdard.ac.in

Reviewer 3 Report

This paper reported on the important vinca alkaloid vincristine produced by a novel Nigrospora zimmermanii isolated from Catharanthus roseus. Through HPTLC and LCMS, only the extracellular filtrate of this fungus was positive for determination of vincristine. Interestingly, an irregular pattern of decrease in vincristine content in subsequent generations was observed. However, the colonization study showed that N. zimmernanii can infect the plant after isolation. Anyway, this method is efficient and easy for obtaining a high yield of vincristine as compared to plant-mediated production. Only minor revisions are required before this manuscript is accepted.

Comments:

1. A trifling question. As shown in Table 1, both Nigrospora camelliae-sinensis CGMCC 3.18125 and Nigrospora pyriformis CGMCC 3.18122 had the same matching degree. Why only considered Nigrospora camelliae-sinensis CGMCC 3.18125 was the next closest homolog?

2. Figure 4 was too blurry.

3. The decrease in vincristine content in subsequent generations was irregular. Was any other vincristine derivatives detected in the culture?

Others:

1. P2L53:  ‘[05]’   ‘[5]’

2. P8L292 & P9L309:  ‘[M+2H]2+ = 413.245’   ‘[M+2H]2+ = 413.2458’

3. P10L334:  ‘Catharanthus roseus’   C. roseus

4. P11L348:  ‘Nigrospora zimmermanii’   N. zimmermanii

5. Please revised reference styles according to the Journal Instructions for Authors. For example, the journal name in [4] was in full name.

Author Response

Reviewers 3

Comments and Suggestions for Authors

This paper reported on the important vinca alkaloid vincristine produced by a novel Nigrospora zimmermanii isolated from Catharanthus roseus. Through HPTLC and LCMS, only the extracellular filtrate of this fungus was positive for determination of vincristine. Interestingly, an irregular pattern of decrease in vincristine content in subsequent generations was observed. However, the colonization study showed that N. zimmernanii can infect the plant after isolation. Anyway, this method is efficient and easy for obtaining a high yield of vincristine as compared to plant-mediated production. Only minor revisions are required before this manuscript is accepted.

Comments:

  1. A trifling question. As shown in Table 1, both Nigrospora camelliae-sinensisCGMCC 3.18125 and Nigrospora pyriformis CGMCC 3.18122 had the same matching degree. Why only considered Nigrospora camelliae-sinensis CGMCC 3.18125 was the next closest homolog?

Ans: Table 1 is revised to supplementary table S1.

The revised table (Supplementary table 1) is prepared as it is generated by NCBI BLASTN programme. Further Sequences producing significant alignments data, generated by NCBI Multiple Sequence Alignment Viewer, Version 1.22.0 is added in the revised supplementary figure S3.

  1. Figure 4 was too blurry.

Ans:  Revised as per the suggestion

  1. The decrease in vincristine content in subsequent generations was irregular. Was any other vincristine derivatives detected in the culture?

Ans: There might be other compounds in the cell extract during subsequent generations. which is not analyzed in our present study. The target compound for the present study was vincristine hence the study was conducted using the specific standard avoiding any other compound detection.

Others:

  1. P2L53:  ‘[05]’  →  ‘[5]’

Ans:  Revised as per the suggestion

  1. P8L292 & P9L309:‘[M+2H]2+ = 413.245’  →  ‘[M+2H]2+ = 413.2458’

Ans:  Revised as per the suggestion

  1. P10L334:‘Catharanthus roseus’  →  ‘ roseus

Ans:  Revised as per the suggestion

  1. P11L348:‘Nigrospora zimmermanii’  →  ‘ zimmermanii

Ans:  Revised as per the suggestion

  1. Please revised reference styles according to the Journal Instructions for Authors. For example, the journal name in [4] was in full name.

Ans:  Revised as per the suggestion

All the authors are thankful to the reviewer for evaluating the article and giving valuable suggestions to improve the article.

Dr. Bibhu Prasad Panda

Email: bppanda@jamiahamdard.ac.in

Round 2

Reviewer 2 Report

Corrections were made and comments were addressed